# IMPROVED SAMPLE COMPLEXITY FOR GLOBAL CONVERGENCE OF ACTOR-CRITIC ALGORITHMS

## ABSTRACT

In this paper, we establish the global convergence of the actor-critic algorithm with a significantly improved sample complexity of $O(\epsilon^{-3})$, advancing beyond the existing local convergence results. Previous works provide local convergence guarantees with a sample complexity of $O(\epsilon^{-2})$ for bounding the squared gradient of the return, which translates to a global sample complexity of $O(\epsilon^{-4})$ using the gradient domination lemma. In contrast to traditional methods that employ decreasing step sizes for both the actor and critic, we demonstrate that a constant step size for the critic is sufficient to ensure convergence in expectation. This key insight reveals that using a decreasing step size for the actor alone is sufficient to handle the noise for both the actor and critic. Our findings provide theoretical support for the practical success of many algorithms that rely on constant step sizes.

## 1 INTRODUCTION

Markov Decision Processes (MDPs) offer a foundational framework for addressing sequential decision-making tasks, with applications spanning diverse fields such as robotics, finance, and healthcare (Sutton & Barto, 2018). The objective of these tasks is to derive a policy that optimally guides decision-making within an environment, relying on local reward signals to inform actions (Puterman, 1994). A variety of approaches have been proposed for discovering such optimal policies or optimizing learning objectives like regret. Beyond value-based methods (Azar et al., 2017; Jin et al., 2018; Agrawal & Agrawal, 2024; Ji & Li, 2024), policy gradient methods have emerged as a prominent and widely adopted class of algorithms (Sutton et al., 2000; Schulman et al., 2017). Gradient-based methods are highly favorable to practice due to their easy adaptability to various policy parametrizations, structured state-action space, and deep learning tools. Despite their extensive empirical success, the theoretical understanding of their convergence properties remains poorly understood until recently. Recent works have explored the global convergence of policy gradient methods using exact gradient oracles across various settings (Agarwal et al., 2020; Mei et al., 2022; Kumar et al., 2024; Liu et al., 2024a;b) with an iteration complexity of $O(\epsilon^{-1})$. However, in practice, the exact gradient is rarely accessible, leading to the adoption of actor-critic methods, where both the actor and critic are updated concurrently using samples from trajectories (Sutton & Barto, 2018; Puterman, 1994).

The theoretical analysis of actor-critic algorithms faces significant challenges due to the strong coupling between the actor and critic. Both components evolve together, influencing each other's updates, with their randomness also entangled. This interaction complicates separating and analyzing them individually. Traditional approaches use two-time-scale methods Borkar (2008), where the actor operates on a slower timescale, treating the critic as nearly converged, and the critic views the actor as nearly stationary Konda & Tsitsiklis (1999); Bhatnagar et al. (2009a). These approaches impose restrictive step sizes, leading to slow convergence.

More recent studies have shifted towards non-asymptotic local convergence in expectation, often measured using the squared norm of the gradient of the return (Zhang et al., 2020b; Olshevsky & Gharesifard, 2023; Chen et al., 2021; Chen & Zhao, 2024; Kumar et al., 2023). These studies achieve a sample complexity of $O(\epsilon^{-2})$, improving the limitations of two-time-scale methods by allowing for less restrictive step sizes. By leveraging gradient domination lemmas, which upper bound the

sub-optimality by the gradient norm, these methods also extend to global convergence, albeit with slower sample complexity rates of $O(\epsilon^{-4})$.

The analysis involves formalizing and analyzing the complex interdependent system (of actor and critic). The key part is to tracking the variance of the critic's Q-value estimation as the policy evolves, establishing its convergence. This simplifies the interdependent system leading to convergence of the actor as well. However, this decaying variance of critic, mandates a decaying step size for the critic, which can't be used for constant step sizes.

To summarize, the existing methods, have huge gap between the policy gradient convergence complexity of $O(\epsilon^{-1})$ and the slower $O(\epsilon^{-4})$ sample-based complexities of actor-critic methods. Additionally, these methods still lack a theoretical explanation for the empirical success observed in practical applications, leaving room for further research into bridging this gap.

n our approach, the critic views the actor's evolving policy as adversarial but treats these changes as diminishing over time, given the actor's decreasing learning rate. Instead of tracking variance as in previous methods, we focus on the critic's bias, proving that this bias decays in line with the actor's learning rate. This allows for any step size in the critic's updates, with constant step sizes being optimal. The key intuition is: In every iterate, the critic reduces the bias while actor increase it by changing the policy. But Q-evaluation being a contraction operator, so it reduces the error by a constant factor while critic ability to mess with critic decreases with time. The outcome being a very efficient tracking by the critic. On the other hand, the actor also operates independently from the critic, only seeing the critic's biased gradient, which diminishes over time with the actor's decreasing learning rate. This effectively decouples the actor and critic while retaining key information.

We formulate a sub-optimality recursion, which is more challenging than in the exact gradient case found in Xiao (2022); Zhang et al. (2020b), due to the presence of a time-dependent learning rate. To address this, we develop an elegant ODE tracking methodology for solving these recursions, yielding significantly improved bounds. Additionally, when this ODE tracking method is applied to the recursion for the exact gradient case, it produces better results compared to existing bounds, such as those in Mei et al. (2022).

**Contributions.** Our contributions are threefolds.

*Firstly*, we establish an improved global convergence of actor-critic methods in softmax-parameterized discounted reward MDPs with sample complexity of $O(\epsilon^{-3})$, hence reducing the gap (see Table 1.1 for a comparison).

*Secondly*, we develop new techniques to show a constant learning rate for the critic is sufficient to ensure global convergence, simplifying algorithmic implementation. This is surprising and defies the traditional wisdom in Borkar (2008); Konda & Tsitsiklis (1999); Bhatnagar et al. (2009a); Dalal et al. (2019); Chen et al. (2021); Chen & Zhao (2024) that proposes use of decreasing step sizes for both the actor and the critic to average out the stochasticity.

*Thirdly*, in the case where the exact gradient is accessible (i.e., policy gradient method), we provide a fine-grained analysis of the convergence rate, demonstrating that even a single iteration improves the current iterate (decreases sub-optimality) (see Table 3). This is unlike the previous results (Mei et al., 2022) where convergence rate was only meaningful when the number of iterations was very high, $k > O\left(\frac{SC_{PL}^2}{(1-\gamma)^5}\right)$[1]. This is made possible due to our new techniques for solving the underlying recurrence relations in the AC algorithm (see Lemma 4), which could be of independent interest.

## 1.1 RELATED WORKS

Our results touch on the convergence of both policy gradient and actor-critic algorithms and we describe the related works to ours below.

**Policy gradient convergence:** Asymptotic convergence of policy gradient has been well-established in Williams (1992); Sutton et al. (1999); Kakade (2001); Baxter & Bartlett (2001). More

---

[1] where $S, \gamma, C_{PL}$ represents the state space cardinality, discount factor and mismatch coefficient respectively

Table 1: Related Work: Sample Complexity of Actor Critic Algorithms

| Work | Convergence | Sample Complexity | Actor Step size $\eta_k$ | Critic Step size $\beta_k$ |
|---|---|---|---|---|
| Konda & Tsitsiklis (1999) | $\|\nabla J^\pi\| \leq \epsilon$ | Asymptotic | $\eta_k = o(\beta_k)$ | $\sum \eta_k^2, \beta_k^2 < \infty = \sum \eta_k, \beta_k$ |
| Bhatnagar et al. (2009a) | $\|\nabla J^\pi\| \leq \epsilon$ | Asymptotic | $\eta_k = o(\beta_k)$ | $\sum \eta_k^2, \beta_k^2 < \infty = \sum \eta_k, \beta_k$ |
| Olshevsky & Gharesifard (2023) | $\|\nabla J^\pi\| \leq \epsilon$ | $O(\epsilon^{-4})$ | $k^{-\frac{1}{2}}$ | $k^{-\frac{1}{2}}$ |
| Chen et al. (2021) | $\|\nabla J^\pi\| \leq \epsilon$ | $O(\epsilon^{-4})$ | $k^{-\frac{1}{2}}$ | $k^{-\frac{1}{2}}$ |
| Chen & Zhao (2024) | $\|\nabla J^\pi\| \leq \epsilon$ | $\tilde{O}(\epsilon^{-4})$ | $k^{-\frac{1}{2}}$ | $k^{-\frac{1}{2}}$ |
| **Ours** | $J^* - J^\pi \leq \epsilon$ | $\tilde{O}(\epsilon^{-3})$ | $k^{-\frac{2}{3}}$ | $\beta$ |

$\tilde{O}$ hides logarthmic factors. Local convergence implies global convergence, see Proposition 1. These works are for different settings such average reward, discounted reward, finite state space, and infinite state space, please refer to the individual work for more details.

recently finite time convergence guarantees were studied in Agarwal et al. (2020); Zhang et al. (2020a); Xu et al. (2020a). Notably Mei et al. (2020) established a key result (gradient domination lemma) that provides sufficient conditions to obtain global convergence guarantees.

**Actor-critic convergence:** The closest to our work is the local convergence of actor-critic algorithm with (equivalent see Table 1) sample complexity of $O(\epsilon^{-4})$ Olshevsky & Gharesifard (2023). This work provides the first sample complexity for the global convergence of the actor-critic algorithm. No additional assumptions are made compared to the setting where the exact gradient is known.

Xu et al. (2020b) establishes the global convergence of the natural actor-critic algorithm with a sample complexity of $O(\epsilon^{-4})$ in discounted reward MDPs. However, the natural actor-critic algorithm demands additional computations, which can be challenging. Yuan et al. (2022) too establishes global convergence with sample complexity of $O(\epsilon^{-3})$, however, it requires an additional structural assumption on the problem which is highly restrictive.

The subsequent paper is organized as follows. In Section 2, we establish key notations, definitions, and results from prior literature. In Section 3, we show how we use our novel solution of recurrence to lead to a fine-grained convergence guarantee for policy gradient (known gradients). Finally in Section 3.1, we describe our results for actor-critic methods

## 2 PRELIMINARIES

We consider the class of infinite horizon discounted reward MDPs with finite state space $\mathcal{S}$ and finite action space $\mathcal{A}$ with discount factor $\gamma \in [0, 1)$ Sutton & Barto (2018); Puterman (1994). The underlying environment is modeled as a probability transition kernel denoted by $P$. We consider the class of randomized policies $\Pi = \{\pi : \mathcal{S} \to \Delta\mathcal{A}\}$, where a policy $\pi$ maps each state to a probability vector over the action space. The transition kernel corresponding to a policy $\pi$ is represented by $P^\pi : \mathcal{S} \to \mathcal{S}$, where $P^\pi(s'|s) = \sum_{a \in \mathcal{A}} \pi(a|s) P(s'|s, a)$ denotes the single step probability of moving from state $s$ to $s'$ under policy $\pi$. Let $R(s, a)$ denote the single step reward obtained by taking action $a \in \mathcal{A}$ in state $s \in \mathcal{S}$. The single-step reward associated with a policy $\pi$ at state $s \in \mathcal{S}$ is defined as $R^\pi(s) = \sum_{a \in \mathcal{A}} \pi(a|s) R(s, a)$. The discounted average reward (or return)

$J^\pi$ associated with a policy $\pi$ is defined as:

$$J^\pi = \mathbb{E}\left[\sum_{n=0}^{\infty} \gamma^n R^\pi(s_k) \mid \pi, P, s_0 \sim \mu\right] = \mu^T(I - \gamma P^\pi)^{-1} R^\pi, \tag{1}$$

where $\mu \in \Delta \mathcal{S}$ denotes the initial state distribution. It can be alternatively expressed as $J^\pi = (1-\gamma)^{-1} \sum_{s \in \mathcal{S}} d^\pi(s) R^\pi(s)$, where $d^\pi = (1-\gamma)\mu^T(I - \gamma P^\pi)^{-1}$ is the stationary measure under the transition kernel $P^\pi$. Value function $v^\pi := (I - \gamma P^\pi)^{-1} R^\pi$ satisfies the following Bellman equation $v^\pi = R^\pi + \gamma P^\pi v^\pi$ Puterman (1994); Bertsekas (2007). The Q-value function $Q^\pi \in \mathbb{R}^{\mathcal{S} \times \mathcal{A}}$ associated with a policy $\pi$ is defined as $Q^\pi(s, a) = R(s, a) + \gamma \sum_{s' \in \mathcal{S}} P(s'|s, a)v^\pi(s')$ for all $(s, a) \in \mathcal{S} \times \mathcal{A}$. For simplicity, we will also assume $\|R\|_\infty \leq 1$.

In this paper, we consider soft-max policy parameterized by $\theta \in \mathbb{R}^{\mathcal{S} \times \mathcal{A}}$ as $\pi_\theta(a|s) = \frac{e^{\theta(s,a)}}{\sum_a e^{\theta(s,a)}}$ Mei et al. (2022). The objective is to obtain an optimal policy $\pi^*$ that maximizes the return $J^\pi$. We denote $J^*$ as a shorthand for the optimal return $J^{\pi^*}$. Among many methods, the policy gradient method is arguably the most widely used algorithm given as

$$\theta_{k+1} := \theta_k + \eta_k \nabla J^{\pi_{\theta_k}}, \tag{2}$$

where $\eta_k$ is the learning rate, in most vanilla form Sutton & Barto (2018). The policy gradient can be derived as

$$\frac{\partial J^{\pi_\theta}}{\partial \theta(s,a)} = (1-\gamma)^{-1} d^{\pi_\theta}(s) \pi_\theta(a|s) A^{\pi_\theta}(s, a),$$

where $A^\pi(s, a) := Q^\pi(s, a) - v^\pi(s)$ is advantage function Mei et al. (2022). The return $J^{\pi_\theta}$ is a highly non-concave function, making global convergence guarantees for the above policy gradient method very challenging. However, the return $J^{\pi_\theta}$ is $L = \frac{8}{(1-\gamma)^3}$-smooth with respect to $\theta$ Mei et al. (2022), leading to the following result.

**Lemma 1.** *[Sufficient Increase Lemma]Mei et al. (2022) With $\eta_k = \frac{1}{L}$ in policy gradient in equation 2, we have the monotone improvement as*

$$J^{\pi_{\theta_{k+1}}} - J^{\pi_{\theta_k}} \geq \frac{1}{2L}\|\nabla J^{\pi_{\theta_k}}\|_2^2, \qquad \forall k \in \mathbb{N}$$

*where $L = \frac{8}{(1-\gamma)^3}$ is smoothness constant of the return $J^{\pi_\theta}$ w.r.t. $\theta$.*

The above result guarantees monotonic improvement at each iteration, ensuring convergence to a stationary point where the gradient becomes zero. Furthermore, the following result provides a crucial structural property of our problem, which is instrumental in achieving global convergence.

**Lemma 2.** *(Performance Difference Lemma, Agarwal et al. (2020)) Let $J^*$ be the globally optimal reward. Then for any $\pi \in \Pi$, the suboptimality of $J^\pi$ can be expressed as:*

$$J^* - J^\pi = (1-\gamma)^{-1} \sum_s d^{\pi^*}(s) \sum_{a \in \mathcal{A}} Q^\pi(s, a)[\pi^*(a|s) - \pi(a|s)].$$

Observe that the result equates the sub-optimality on RHS (a global quantity) with only the Q-value (a local quantity). This is crucially exploited in Gradient Domination Lemma 3 that upper bounds the sub-optimality with the norm of the gradient.

**Lemma 3.** *(Gradient Domination Lemma, Mei et al. (2022)) The sub-optimality is upper bounded by the norm of the gradient as*

$$\|\nabla J^{\pi_{\theta_k}}\|_2 \geq \frac{c}{\sqrt{S} C_{PL}}\left[J^* - J^{\pi_{\theta_k}}\right],$$

*where $C_{PL} = \max_k \|\frac{d^{\pi^*}}{d^{\pi_{\theta_k}}}\|_\infty$ is mismatch coefficient and $c = \min_k \min_s \pi_{\theta_k}(a^*(s)|s)$,*

The result states that the norm of the gradient vanishes only when the sub-optimality is zero. In other words, the gradient is zero only at the optimal policies. This, combined with the Sufficient Increase Lemma, directly leads to the global convergence of the policy gradient update rule in equation 2.

However, the above lemma requires the mismatch coefficient $C_{PL}$ to be bounded, which can be ensured by setting the initial distribution $\mu(s) > 0$ for all states. Additionally, the result requires the constant $c$ to be strictly greater than zero. This condition can be satisfied by initializing the parameterization with $\theta_0 = 0$ or by ensuring it remains bounded. Furthermore, as the iterates progress towards an optimal policy, the constant $c$ remains bounded away from zero.

## 3 MAIN

The policy gradient method is widely regarded as one of the most successful algorithms in practice, with numerous variants developed over the past few decades Sutton & Barto (2018); Schulman et al. (2015); Mnih et al. (2015). However, its theoretical convergence properties remained poorly understood until recent works Bhandari & Russo (2024); Agarwal et al. (2020); Xiao (2022); Mei et al. (2022); Kumar et al. (2024) established global convergence guarantees for equation 2, where $\eta_k = \frac{1}{L}$ is the learning rate, with $L = \frac{8}{(1-\gamma)^3}$ being the smoothness constant of the return $J^{\pi_\theta}$ with respect to $\theta$ Mei et al. (2022).

It is standard to equate the gradient $(\|\nabla J^{\pi_{\theta_k}}\|_2)$ in the Gradient Domination Lemma 3 and the Sufficient Increase Lemma 1, leading to:

$$2L \left( J^{\pi_{\theta_{k+1}}} - J^{\pi_{\theta_k}} \right) \geq \frac{c^2}{SC_{PL}^2} \left[ J^* - J^{\pi_{\theta_k}} \right]^2,$$

which allows us to obtain the following sub-optimality recursion, with $a_k := J^* - J^{\pi_{\theta_k}}$ and $\sigma = \frac{c^2(1-\gamma)^3}{16SC_{PL}^2}$:

$$a_k - a_{k+1} \geq \sigma a_k^2. \tag{3}$$

The result below provides the bound on the above recursion. Note that proof of all the results in the paper, can be found in the appendix.

**Lemma 4.** *Given $a_k - a_{k+1} \geq \sigma a_k^2$, where $\sigma = \frac{c^2(1-\gamma)^3}{16SC_{PL}^2}$, we have*

$$a_k \leq \frac{1}{\frac{1}{a_0} + \sigma k}, \qquad \forall k \geq 0.$$

The proof is provided in Appendix A.1. Lemma 4 directly yields the convergence result below.

**Theorem 1.** *The exact policy gradient iterates equation 2, with learning rate $\eta_k = \frac{1}{L}$ and $\sigma = \frac{c^2(1-\gamma)^3}{16SC_{PL}^2}$, converge as*

$$J^* - J^{\pi_{\theta_k}} \leq \frac{1}{\frac{1}{J^* - J^{\pi_{\theta_0}}} + \sigma k}.$$

Asymptotically (when $k \to \infty$), the above rate is approximately $\frac{1}{\frac{1}{J^* - J^{\pi_{\theta_0}}} + \sigma k}$, which is similar to the existing rate of $\frac{16SC_{PL}^2}{c^2(1-\gamma)^6 k}$ Mei et al. (2022), with the improvement by a factor of $\frac{1}{(1-\gamma)^3}$. However, note that the constant $\frac{16SC_{PL}^2}{c^2(1-\gamma)^6} \gg \frac{2}{1-\gamma}$ is significantly larger than the worst possible sub-optimality of $\frac{2}{1-\gamma}$. Therefore, the existing rate $\frac{16SC_{PL}^2}{c^2(1-\gamma)^6 k}$ does not provide meaningful bounds for initial iterates until $k = \frac{8SC_{PL}^2}{c^2(1-\gamma)^5}$. In contrast, our result improves the existing rate by a factor of $\frac{1}{(1-\gamma)^3}$ for large $k \gg 1$ and offers meaningful convergence even for small $k$.

Table 2: Relative sub-optimality ($\frac{1-\gamma}{2} a_k$) with iterates

| $k$ | 0 | 1 | 10 | $10^2$ | $10^3$ | $10^4$ | $10^5$ | $10^6$ | $10^7$ |
|---|---|---|---|---|---|---|---|---|---|
| Mei et al. (2022) | $\infty$ | $8*10^7$ | $8*10^6$ | $8*10^5$ | $8*10^4$ | 8000 | 800 | 80 | 8 |
| **Ours** | 1 | 0.999998 | 0.99998 | 0.9998 | 0.998 | 0.98 | 0.88 | 0.44 | 0.07 |

Taking $\gamma = 0.9, S = 1000, C_{PL} = c = 1, J^* - J_0 = \frac{2}{1-\gamma}$

The difference between our bound and the previous one is illustrated numerically in Table 2, where $\frac{1}{\frac{1}{J^* - J^{\pi_{\theta_0}}} + \sigma k}$ and $\frac{1}{\frac{1}{J^* - J^{\pi_{\theta_0}}} + \sigma k}$ are computed for different $k$. In Table 2, it can be seen that the previous bounds provide a significantly large upper bound on the sub-optimality for the initial iterates, while our method demonstrates improvement from the very first iterates.

The recursion $a_k - a_{k+1} \geq \sigma a_k^2$ occurs in the projected policy gradient update rule (with direct parameterization) in Liu et al. (2024b;a), yielding a convergence rate of $O\left(\frac{\sigma}{k}\right)$. Consequently, Lemma 4 offers a similar result improvement, resulting in a convergence rate of $O\left(\frac{1}{\frac{1}{a_0} + \sigma k}\right)$.

Now we move to the analysis of actor-critic algorithm which is the core contribution of this work.

### 3.1 ACTOR-CRITIC METHODS

Above studies, however, assume access to the exact policy gradient at each iteration, which is rarely feasible in practice. In this work, we focus on sample based policy gradient method also known as actor critic algorithm. The actor-critic algorithm, which is sample-based policy gradient method and thus more practical, presents a greater challenge. Actor-critic methods have been studied for a long time, from asymptotic convergence on two timescales Konda & Tsitsiklis (1999); Bhatnagar et al. (2009b) to finite time two time scale Zhang et al. (2020b); Olshevsky & Gharesifard (2023); Chen et al. (2021) until finite-time single-timescale convergence Chen & Zhao (2024). These works establish local convergence bounding the average expected square of gradient of the return, with following state-of-the-art rate

$$\sum_{k=1}^{K} \frac{1}{K} \|\nabla J^{\pi_k}\|_2^2 \leq O(K^{-\frac{1}{2}}), \qquad Chen\&Zhao \; (2024).$$

Using Gradient Domination Lemma 3, this local convergence translates to $O(\epsilon^{-4})$ sample complexity of global convergence, as shown in the result below.

**Proposition 1.** *If $E\|\nabla J^{\pi_k}\|_2^2 \leq O(k^{-\frac{1}{2}})$ then $J^* - EJ^{\pi_k} \leq O(k^{-\frac{1}{4}})$.*

*Proof.* From Gradient Domination Lemma 3, we have

$$E\|\nabla J^{\pi_k}\|_2^2 \geq E\left[\; J^* - J^{\pi_k}\;\right] \geq \frac{c^2}{SC_{PL}^2}\left[\; J^* - EJ^{\pi_k}\;\right]^2, \qquad \text{(from Jenson's inequality).}$$

Hence if $E\|\nabla J^{\pi_k}\|_2^2 \leq O(k^{-\frac{1}{2}})$ then $\left[\; J^* - EJ^{\pi_k}\;\right]^2 \leq O(k^{-\frac{1}{2}})$, implying $J^* - EJ^{\pi_k} \leq O(k^{-\frac{1}{4}})$. $\qquad \square$

---

**Algorithm 1** Online Actor Critic Algorithm

---

**Input**: Initialize $Q_0$ and $\theta_0$ arbitrarily, and $\eta_k = \eta_0 \left(\frac{1}{1+c_6 k}\right)^{\frac{2}{3}}$ for $k \geq 0$, where the $c_6$ is a positive constant.

1: **while** not converged; $k = k + 1$ **do**
2:     Sample $(s,a) \sim d^{\pi_{\theta_k}}$ and get the next state and action $s' \sim P(\cdot|s,a), a' \sim \pi_{\theta_k}(\cdot|s')$ .

3:     Update the policy parameter

$$\theta_{k+1}(s,a) = \theta_k(s,a) + \eta_k(1-\gamma)^{-1} A_k(s,a),$$

    where $A_k(s,a) = Q_k(s,a) - v_k(s)$ and $v_k(s) = \sum_a \pi_{\theta_k}(a|s)Q_k(s,a)$.

4:     Update Q-value

$$Q_{k+1}(s,a) = Q_k(s,a) + \beta\left[\; R(s,a) + \gamma Q_k(s',a') - Q_k(s,a)\;\right].$$

5: **end while**

---

In this work, we study a simple actor-critic algorithm in Algorithm 1. Our objective is to obtain a policy $\pi$ that maximizes the expected discounted return $J^\pi$ using the samples. Since, the Algorithm 1 is random, hence we focus on the expected return $J_k := E[J^{\pi_{\theta_k}}]$ at time $k$.

Note that the algorithm requires samples $s_k \sim d^{\pi_{\theta_k}}$ at each iteration, which is a common assumption in most works on the discounted reward setting Zhang et al. (2020b); Konda & Tsitsiklis (1999); Bhatnagar et al. (2009a); Chen et al. (2021); Kumar et al. (2023); Olshevsky & Gharesifard (2023); Chen & Zhao (2024). This can be achieved by initializing the Markov chain with $s_0 \sim \mu$, and at each step $i$, continuing the chain with probability $\gamma$ by sampling $s_{i+1} \sim P^{\pi_{\theta_k}}(\cdot|s_i)$, or terminating the chain with probability $(1 - \gamma)$. Once the chain terminates, we randomly select a state uniformly as $s_k$. This process ensures that the state $s_k$ is sampled from $d^{\pi_{\theta_k}}$. However, this approach increases the average computational complexity by a factor of $\frac{1}{1-\gamma}$. There may be more efficient methods to achieve this sampling, but we omit these for clarity.

## 3.2 CRITIC CONVERGENCE

In this section, we analyze the expected convergence of the Q-value evaluation (critic) in Algorithm 1, using samples drawn from the evolving policy, which is complex. Therefore, we first fix a policy $\pi$ and consider the Q-value evaluation:

$$Q_{k+1}(s_k, a_k) = Q_k(s_k, a_k) + \beta \left[ R(s_k, a_k) + \gamma Q_k(s'_k, a'_k) - Q_k(s_k, a_k) \right], \qquad (4)$$

where the states are $s_k \sim d^\pi$, the actions are $a_k \sim \pi(\cdot|s_k)$, the next states are $s'_k \sim P(\cdot|s_k, a_k)$, and the subsequent actions are $a'_k \sim \pi(\cdot|s'_k)$. Taking the expectation , we obtain:

$$\mathbb{E}Q_{k+1} = \mathbb{E}Q_k + \beta D^\pi \left[ R + \gamma P_\pi \mathbb{E}Q_k - \mathbb{E}Q_k \right],$$

where $D^\pi((s,a),(s',a')) = d^\pi(s)\pi(a|s)\mathbf{1}((s,a) = (s',a'))$ is a diagonal matrix, and $P_\pi((s',a'),(s,a)) = P(s'|s,a)\pi(a'|s')$ represents the transition dynamics under policy $\pi$.

Lets define momentum Bellman operator as

$$T_\beta^\pi Q := Q + \beta D^\pi (R + \gamma P_\pi Q - Q),$$

then observe that $T_\beta^\pi Q^\pi = Q^\pi$. To ensure the convergence of $\mathbb{E}Q_k$, we make **one of the following three** assumptions.

**Assumption 1.** *We assume that one of the following holds for all $Q, \pi$*

1. *There exists a $\lambda > 0$ such that:*

$$\langle Q^\pi - Q, D^\pi(I - \gamma P_\pi)Q^\pi - Q \rangle \geq \lambda \|Q^\pi - Q\|_2^2,$$

   *which implies: $\|T_\beta^\pi Q - Q^\pi\|_2 \leq \alpha \|Q - Q^\pi\|_2$, (from Proposition 5), where $\alpha = \sqrt{1 - \frac{\lambda^2}{2}}$ taking $\beta = \frac{\lambda}{2}$.*

2. *There exists an $\alpha < 1$ such that $\|D^\pi(T_\beta^\pi Q - Q^\pi)\|_\infty \leq \alpha \|D^\pi(Q - Q^\pi)\|_\infty$.*

3. *There exists an $\alpha < 1$ such that $\|D^\pi(L_\beta^\pi A - A^\pi)\|_\infty \leq \alpha \|D^\pi(A - A^\pi)\|_\infty$, where $A^\pi(s,a) = Q^\pi(s,a) - \sum_a \pi(a|s)Q^\pi(s,a)$, and $(L_\beta^\pi A)(s,a) = (T_\beta^\pi Q)(s,a) - \sum_a \pi(a|s)(T_\beta^\pi Q)(s,a)$.*

The first assumption is standard and has been made in all the previous works, to the best of our knowledge Olshevsky & Gharesifard (2023); Chen et al. (2021); Chen & Zhao (2024); Bhatnagar et al. (2009a); Konda & Tsitsiklis (1999); Zhang et al. (2020b). It guarantees the convergence of Q-value evaluation. Specifically, under the first item of Assumption 1, the update rule equation 4 converges as $\|\mathbb{E}Q_k - Q^\pi\|_2 \to \alpha^k \|\mathbb{E}Q_0 - Q^\pi\|_2$. Unfortunately this assumption is too restrictive as it requires $d^\pi(s,a) \geq 0$ for all state and action, as the co-ordinates which are not visited by the policy, can't be updated.

Fortunately, in policy gradient methods, we don't require $Q^\pi$ instead $D^\pi Q^\pi$ for the gradient updates, hence, the second item in the Assumption 1 is enough to ensure $\|D^\pi(\mathbb{E}Q_k - Q^\pi)\|_\infty \to \alpha^k \|\mathbb{E}Q_0 - Q^\pi\|_\infty$ in the Q-value evaluation update rule equation 4.

Further, the last item in Assumption 1, is even less restrictive than the second item. As the policy update progress in the actor-critic method, the policy $\pi_k$ converges to a deterministic optimal policy consequently $A^{\pi_k} \to 0$ ( as $A^\pi = 0$ for deterministic policy $\pi$). This may leads to even better error control in the critic updates.

**Q-evaluation with evolving policy with coupled randomness.** The above assumption ensures the convergence gradient evaluation for the fixed policy. In our case (Algorithm 1) the policy $\pi_{\theta_k}$ evolves but with decreasing rate ($\|\theta_{i+1} = \theta_i\| \leq \frac{2}{(1-\gamma)^2}$ as $|A_{k+1}(s, a)| \leq \frac{2}{1-\gamma}$). Hence, we have the following Q-value evaluation:

$$Q_{k+1}(s_k, a_k) = Q_k(s_k, a_k) + \beta \left[ R(s_k, a_k) + \gamma Q_k(s'_k, a'_k) - Q_k(s_k, a_k) \right], \tag{5}$$

where $s_k \sim d^\pi, a_k \sim \pi^{\theta_k}(\cdot|s_k), s'_k \sim P(\cdot|s_k, a_k)$, and $a'_k \sim \pi^{\theta_k}(\cdot|s'_k)$. Moreover, $\{\theta_k\}_{k\geq 0}$ and $\{s_k\}_{k\geq 0}$ are not independent, that is, their noise are coupled. Hence, $E[Q_k] = E_{\{\theta_i\}_{k\geq 0}}[\Pi_{i=0}^{k-1} T_\beta^{\pi_{\theta_i}} Q_0]$ is not satisfied.

This coupling makes the analysis of the bias term $E\langle D^{\pi_{\theta_k}} A^{\pi_{\theta_k}}, D^{\pi_{\theta_k}} (A_k - A^{\pi_{\theta_k}}) \rangle$ very challenging (this term is required in the actor section) and a core technical contribution of the paper. We use novel techniques in the result below computing the term.

The result below states that each of three assumptions above can ensure the evaluation of the gradient under the evolving policy.

**Lemma 5.** *In Algorithm 1, we have*

$$|E\langle D^{\pi_{\theta_k}} A^{\pi_{\theta_k}}, D^{\pi_{\theta_k}} (A_k - A^{\pi_{\theta_k}}) \rangle| \leq c_q \eta_k, \qquad \forall k \geq 1.$$

*where $c_q$ is a constant defined in the appendix.*

The above result presented in a form which convenient for the analysis later, however it essentially implies the bias in the gradient diminishes over time that is $\|ED^{\pi_{\theta_k}}(A_k - A^{\pi_{\theta_k}})\|_\infty = O(\eta_k)$.

The proof of the above result can be found in the appendix. We know that Q-value evaluation is $\gamma$-contraction operator Sutton & Barto (2018); Puterman (1994), that is, Q-evaluation converges linearly for the fixed policy. The above result states that the for the evolving policy, Q-evaluation tracking is also very efficient. Notably, the result above ensures the convergence of critic in expectation irrespective of actor. This greatly helps in the analysis of the Algorithm 1, as it decouples the critic and actor.

### 3.3 CONVERGENCE OF ACTOR

In this section, we focus on the convergence of the actor update rule,

$$\theta_{k+1}(s_k, a_k) = \theta_k(s_k, a_k) + \eta_k(1 - \gamma)^{-1} A_{k+1}(s_k, a_k), \tag{6}$$

as presented in Algorithm 1. Note that by construction, $\|A_k\|_\infty \leq \frac{2}{1-\gamma}$, a fact that will be used in the analysis later.

We begin with deriving a sufficient increase lemma for our noisy and biased gradient ascent which can be seen as the extension of the similar result in Mei et al. (2022).

**Lemma 6.** *Let $\theta_k$ be the iterate obtained Algorithm 1 . Then,*

$$E[J^{\pi_{\theta_{k+1}}} - J^{\pi_{\theta_k}}] \geq \frac{\eta_k}{1-\gamma} \left[ E\|\nabla J^{\pi_{\theta_k}}\|_2^2 + E\langle D^{\pi_{\theta_k}} A^{\pi_{\theta_k}}, D^{\pi_{\theta_k}} (A_k - A^{\pi_{\theta_k}}) \rangle - \frac{2L\eta_k}{(1-\gamma)^3} \right].$$

The proof in the appendix.

Recall that we use $J_k := EJ^{\pi_{\theta_k}}$ as a shorthand. The result below provides the sub-optimality recursion.

**Lemma 7.** *Taking $a_k = \frac{1-\gamma}{2}(J^* - J_k)$, we get the recursion*

$$a_k - a_{k+1} \geq \eta_k \left[ c_1 a_k^2 - c_2 \eta_k \right], \tag{7}$$

*where $c_2 := \frac{c_q}{1-\gamma} + \frac{L}{(1-\gamma)^3}, c_1 := \frac{c^2}{2SC_{PL}^2}, L = \frac{8}{(1-\gamma)^3}$ smoothness coefficient, and the constant $c_q$ is defined in the appendix.*

**Lemma 8.** *The $a_k$ is upper bounded as*

$$a_k \leq \frac{1}{(1 + c_6 k)^{\frac{1}{3}}}, \qquad \forall k \geq 0,$$

*where $c_6 = \frac{3c_1^2}{4c_2}$.*

Upper bounding the above recursion is very challenging (and a core technical contribution of the paper) due presence of time dependent variables. We develop elegant methods to solve this recursion which can be found in the appendix.

### 3.4 ACTOR-CRITIC CONVERGENCE

The result below demonstrates the convergence of Algorithm 1 with a sample complexity of $O(\epsilon^{-3})$, which is significantly faster than the existing sample complexity of $O(\epsilon^{-4})$, as summarized in Table 1.

**Theorem 2.** *For step size* $\eta_k = \eta_0 \left( \frac{1}{1+c_6 k} \right)^{\frac{2}{3}}, \beta_k = \beta$ *in Algorithm 1, we have the following convergence*

$$J^* - J_k \leq \frac{2}{(1-\gamma)(1+c_6 k)^{\frac{1}{3}}}, \qquad \forall k \geq 0,$$

*where* $c_6 = \frac{3c^4(1-\gamma)^2}{16S^2 C_{PL}^4 (\frac{c_q}{1-\gamma} + \frac{L}{(1-\gamma)^3})}$.

We summarize the key components of the result as:

1. **Convergence Rate of Q-Value Evaluation with Diminishing Adversarial Evolving Policy** : Since Q-value evaluation is a $\gamma$-contraction for a fixed policy, we find that with a slowly changing policy, the Q-value evaluation converges at the same rate as the policy itself.

2. **Convergence of Actor using Gradient with Diminishing Bias**: We obtain sub-optimality recursion using modified sufficient increase lemma derived from smoothness of the return and gradient domination lemma.

3. **Solving the Recursion**: Solving the recursion is the most challenging part of this work. We develop a general framework that demonstrates how this recursion mirrors the behavior of the underlying ordinary differential equation (o.d.e.).

## 4 CONCLUSION AND DISCUSSION

We establish global convergence of actor-critic algorithms with significantly improved sample complexity of $O\epsilon^{-3}$ compared to existing rate of $O(\epsilon^{-4})$.The most remarkable finding of our analysis is the guaranteed convergence of the algorithm with a constant learning rate for the critic, as often used in practice. Traditionally, decreasing step sizes are deemed essential for both the actor and critic to reduce noise Borkar (2008); Olshevsky & Gharesifard (2023); Chen et al. (2021); Chen & Zhao (2024). However, our results show that a decreasing step size for the actor alone suffices to average out noise, marking a departure from the two-time-scale theory. This works reduces the gap between theoretical understanding and empirical success of the algorithm.

Our framework is quite general, hence the approach can be extended to other settings such as average reward, function approximation setting. We leave this for the future work.

In traditional two-time-scale algorithms, the inner loop perceives the outer loop as stationary, while the outer loop regards the inner loop as converged, leading to asymptotic decoupling Borkar (2008).

We introduce a novel analytical framework where the inner and outer loops are "blind" to each other. The the inner loop (critic) views the the outer loop (actor) as adversarial but with diminishing influence as the actor's learning rate decreases, leading to the critic's bias reduction over time. On the other hand, the outer loop, uses this critic biased feedback (gradient) provided by the inner loop. Since, the bias diminishes with time, it allows the outer loop converge.

This framework for resolving the two time scale coupling, combined with our novel elegant methodology to bound the recursions, can be used to analysis other two-time scale algorithms.

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

## A   Proof of Main Text

### A.1   Exact Gradient Policy Gradient

**Lemma 9.** *Given $a_k - a_{k+1} \geq \sigma a_k^2$, where $\sigma = \frac{c^2(1-\gamma)^3}{16SC_{PL}^2}$, we have*

$$a_k \leq \frac{1}{\frac{1}{a_0} + \sigma k}, \qquad \forall k \geq 0.$$

*Proof.* Let $u_k := \frac{1}{\frac{1}{a_0} + \sigma k}$, implying $\frac{du_k}{dk} = -\sigma u_k^2$. We have $\sigma \leq \frac{1-\gamma}{4}$ as $C_{PL}, S \geq 1, c \leq 1$ by definition, this implies $\frac{1}{2\sigma} \geq \frac{2}{1-\gamma} \geq a_0 \geq u_k$. We will prove by induction that $a_k \leq u_k$ for all $k \geq 0$. For the base case, we have $a_0 \leq u_0 = a_0$, by definition. Assuming $a_k \leq u_k$, we have

$$a_{k+1} \leq a_k - \sigma a_k^2, \qquad \text{(from definition } \sigma \leq \sigma\text{)},$$

$$\leq u_k - \sigma u_k^2, \qquad \text{(as } h(x) = x - \sigma x^2 \text{ is increasing for } x \leq \frac{1}{2\sigma} \text{ and } a_k \leq u_k \leq a_0 \leq \frac{1}{2\sigma}\text{)},$$

$$= u_k - \sigma \int_{x=k}^{k+1} u_k^2 dx, \qquad \text{(dummy integral)},$$

$$\leq u_k - \sigma \int_{x=k}^{k+1} u_x^2 dx, \qquad \text{(as } u_x \text{ is decreasing with } x\text{)},$$

$$= u_{k+1}, \qquad \text{(as } \frac{du_k}{dk} = -\sigma u_k^2 \text{ )}.$$

This proves $a_k \leq u_k \leq \frac{1}{\frac{1}{a_0} + \sigma k}$ for all $k \geq 0$. $\qquad\square$

### A.2   Convergence of Critic

In this section, we take Assumption 1, item 2. For the item 1 and item 3, similar proof and intuitions follows.

**Proposition 2.** *Given $\|Q\|_\infty \leq \frac{1}{1-\gamma}$, we have,*

$$|\langle D^{\pi_1} A^{\pi_2}, D^{\pi} \left( (T_\beta^{\pi_3})^m Q - Q^{\pi_3} \right) \rangle| \leq \frac{4\alpha^m}{(1-\gamma)^2}, \qquad \forall \pi_i, m.$$

*Proof.*

$$|\langle D^{\pi_1} A^{\pi_2}, D^{\pi} \left( (T_\beta^{\pi_3})^m Q - Q^{\pi_3} \right) \rangle| \leq \|D^{\pi_1} A^{\pi_2}\|_1 \|D^{\pi_3}(T_\beta^{\pi_3})^m (Q - Q^{\pi_3})\|_\infty$$

$$\leq \frac{2}{1-\gamma} \|D^{\pi_3} \left( (T_\beta^{\pi_3})^m Q - Q^{\pi_3} \right)\|_\infty, \qquad \text{(as } \|A^\pi\|_\infty \leq \frac{2}{1-\gamma}, \|d^\pi\|_1 = 1\text{)}$$

$$\overset{(a)}{\leq} \frac{2\alpha^m}{1-\gamma} \|D^{\pi_3}(Q - Q^{\pi_3})\|_\infty$$

$$\leq \frac{4\alpha^m}{(1-\gamma)^2}, \qquad \text{(as } \|Q\|_\infty, \|Q^\pi\|_\infty \leq \frac{1}{1-\gamma}\text{)}$$

The inequality (a) comes from recursivelyusing $\|D^\pi(T_\beta^\pi Q - Q^\pi)\|_\infty \leq \alpha \|D^\pi(Q - Q^\pi)\|_\infty$, as assumed in the Assumption 1, item 2

$\square$

**Lemma 10.** *In Algorithm 1, we have*

$$|E\langle D^{\pi_{\theta_k}} A^{\pi_{\theta_k}}, D^{\pi_{\theta_k}} (A_k - A^{\pi_{\theta_k}}) \rangle| \leq C_q \eta_k, \qquad \forall k \geq 1.$$

*Proof.* We show our proof works under the Assumption 1, item 2. For item 1 and item 3, similar proof follows.

Lets fix $k$. Lets deine the remainder (smoothness) term: For $0 \leq m \leq k-1$,

$$c_{k-m-1} := E\langle D^{\pi_{\theta_{k-m}}} A^{\pi_{\theta_{k-m}}}, D^{\pi_{\theta_{k-m}}}(\frac{1}{\alpha}T_\beta^{\pi_{\theta_{k-m}}})^m \left( Q_{k-m} - Q^{\pi_{\theta_{k-m}}} \right)\rangle$$

$$- E\langle D^{\pi_{\theta_{k-m-1}}} A^{\pi_{\theta_{k-m-1}}}, D^{\pi_{\theta_{k-m-1}}}(\frac{1}{\alpha}T_\beta^{\pi_{\theta_{k-m-1}}})^m \left( Q_{k-m} - Q^{\pi_{\theta_{k-m-1}}} \right)\rangle$$

$$= E\langle D^{\pi_{\theta_{k-m}}} A^{\pi_{\theta_{k-m}}}, D^{\pi_{\theta_{k-m}}}(\frac{1}{\alpha}G_\beta^{\pi_{\theta_{k-m}}})^m \left( Q_{k-m} - Q^{\pi_{\theta_{k-m}}} \right)\rangle$$

$$- E\langle D^{\pi_{\theta_{k-m-1}}} A^{\pi_{\theta_{k-m-1}}}, D^{\pi_{\theta_{k-m-1}}}(\frac{1}{\alpha}G_\beta^{\pi_{\theta_{k-m-1}}})^m \left( Q_{k-m} - Q^{\pi_{\theta_{k-m-1}}} \right)\rangle,$$

where $G^\pi := (I - \beta D^\pi(I - \gamma P_\pi))$. From the Assumption 1, item 2, we have $\|D^\pi G^\pi v\|_\infty \leq \alpha\|D^\pi v\|_\infty$ for all $v$. This implies the operator norm of $\frac{1}{\alpha}D^\pi G^\pi v$ is 1. This implies all the consitituents term in $C_{k-m-1}$ have norm $O(1)$ (that is no dependence on $\alpha$). Hence, from smoothness, we get $|c_k| \leq C\eta_{k-1}$ for some constant $C$ dependent only possibly on $\frac{1}{1-\gamma}$, $S$ and $A$.

Now, we focus on the original term. By definition, we have

$$E\langle D^{\pi_{\theta_k}} A^{\pi_{\theta_k}}, D^{\pi_{\theta_k}}(Q_k - Q^{\pi_{\theta_k}})\rangle$$

$$= c_{k-1} + E\langle D^{\pi_{\theta_{k-1}}} A^{\pi_{\theta_{k-1}}}, D^{\pi_{\theta_{k-1}}}(Q_k - Q^{\pi_{\theta_{k-1}}})\rangle$$

$$= c_{k-1} + E\left[ E\left[ \langle D^{\pi_{\theta_{k-1}}} A^{\pi_{\theta_{k-1}}}, D^{\pi_{\theta_{k-1}}}(Q_k - Q^{\pi_{\theta_{k-1}}})\rangle \mid \mathcal{F}_{k-1} \right] \right],$$

$$= c_{k-1} + E\left[ \langle D^{\pi_{\theta_{k-1}}} A^{\pi_{\theta_{k-1}}}, D^{\pi_{\theta_{k-1}}}\left( E\left[ Q_k \mid \mathcal{F}_{k-1} \right] - Q^{\pi_{\theta_{k-1}}} \right)\rangle \right]$$

$$= c_{k-1} + E\left[ \langle D^{\pi_{\theta_{k-1}}} A^{\pi_{\theta_{k-1}}}, D^{\pi_{\theta_{k-1}}}\left( T_\beta^{\pi_{\theta_{k-1}}} Q_{k-1} - Q^{\pi_{\theta_{k-1}}} \right)\rangle \right], \qquad \text{(from def. of } T_\beta^\pi)$$

$$= c_{k-1} + E\left[ \langle D^{\pi_{\theta_{k-1}}} A^{\pi_{\theta_{k-1}}}, D^{\pi_{\theta_{k-1}}} T_\beta^{\pi_{\theta_{k-1}}}\left( Q_{k-1} - Q^{\pi_{\theta_{k-1}}} \right)\rangle \right], \qquad \text{(as } T_\beta^\pi Q^\pi = Q^\pi)$$

$$= c_{k-1} + \alpha E\left[ \langle D^{\pi_{\theta_{k-1}}} A^{\pi_{\theta_{k-1}}}, D^{\pi_{\theta_{k-1}}}\frac{1}{\alpha}T_\beta^{\pi_{\theta_{k-1}}}\left( Q_{k-1} - Q^{\pi_{\theta_{k-1}}} \right)\rangle \right], \qquad \text{(note } T_\beta^\pi \text{ is assumed } \alpha \text{ contraction)}$$

$$= c_{k-1} + \alpha c_{k-2} + \alpha^2 E\left[ \langle D^{\pi_{\theta_{k-2}}} A^{\pi_{\theta_{k-2}}}, D^{\pi_{\theta_{k-2}}}(\frac{1}{\alpha}T_\beta^{\pi_{\theta_{k-2}}})^2\left( Q_{k-1} - Q^{\pi_{\theta_{k-2}}} \right)\rangle \right], \qquad \text{(unrolling one more step)}$$

$$= \sum_{m=0}^{k-1} \alpha^{k-1-m}c_m + \alpha^k E\left[ \langle D^{\pi_{\theta_0}} A^{\pi_{\theta_0}}, D^{\pi_{\theta_0}}(\frac{1}{\alpha}T_\beta^{\pi_{\theta_0}})^k\left( Q_0 - Q^{\pi_{\theta_0}} \right)\rangle \right], \qquad \text{(unrolling till end)}$$

Now, taking the absolute value, and using triangle inequlaity, we have

$$|E\langle D^{\pi_{\theta_k}} A^{\pi_{\theta_k}}, D^{\pi_{\theta_k}}(Q_k - Q^{\pi_{\theta_k}})\rangle| \tag{8}$$

$$\leq \sum_{m=1}^{k} \alpha^{m-1}|c_{k-m}| + \frac{4\alpha^k}{(1-\gamma)^2}, \qquad \text{(from Proposition 2),} \tag{9}$$

$$\leq C\sum_{m=1}^{k} \alpha^{m-1}\eta_{k-m} + \frac{4\alpha^k}{(1-\gamma)^2} \tag{10}$$

$$\leq Cc_\eta\eta_k, \qquad \text{(from Proposition 3).} \tag{11}$$

Similarly, we can bounds on

$$|E\langle D^{\pi_{\theta_k}} A^{\pi_{\theta_k}}, D^{\pi_{\theta_k}}(Q_k - Q^{\pi_{\theta_k}})\rangle| \leq c_q\eta_k,$$

for appropriate $c_q$.

$\square$

We know that Q-value evaluation is $\gamma$-contraction operator Sutton & Barto (2018); Puterman (1994), that is, Q-evaluation converges linearly for the fixed policy. The above result states that the for the evolving policy, Q-evaluation tracking is also very efficient. Notably, the result above ensures the convergence of critic in expectation irrespective of actor. This greatly helps in the analysis of the Algorithm 1, as it decouples the critic and actor.

**Proposition 3** (Critic Bound). *Given $\eta_k = \eta_0 u_k^2$, we have*

$$\sum_{k=0}^{N-1} \alpha^{N-k} \eta_k + \kappa \alpha^{N+1} \leq c_\alpha \eta_N, \qquad \forall N \geq 0$$

*where $u_k = \left( \frac{1}{1+ck} \right)^{\frac{1}{3}}, 0 \leq \alpha, c < 1$ and $c_\eta = \max_{N \geq 0} \left( 2\alpha^{\frac{N}{2}} \left( 1 + \frac{N}{2} \right)^{\frac{5}{3}} + \frac{2}{1-\alpha} + \alpha^{N+1}(1 + cN)^{\frac{1}{3}} \frac{\kappa}{\eta_0} \right)$.*

*Proof.* We have

$$\sum_{k=0}^{N-1} \alpha^{N-k} u_k^2 = \sum_{k=0}^{M} \alpha^{N-k} u_k^2 + \sum_{k=M+1}^{N-1} \alpha^{N-k} u_k^2, \tag{12}$$

$$\leq \sum_{k=0}^{M} \alpha^{N-M} u_k^2 + \sum_{k=M+1}^{N-1} \alpha^{N-k} u_M^2, \qquad \text{(as } u_k^2, \alpha^k \text{ are decreasing with } k) \tag{13}$$

$$\leq \sum_{k=0}^{M} \alpha^{N-M} u_k^2 + \frac{u_M^2}{1-\alpha}, \qquad \text{(geometric sum)} \tag{14}$$

$$\leq \alpha^{N-M}(M+1) + \frac{u_M^2}{1-\alpha}, \qquad \text{(as } u_k^2 \leq 1) \tag{15}$$

$$= \left( 1 + \frac{cN}{2} \right)^{-\frac{2}{3}} \left( \frac{\alpha^{\frac{N}{2}} \left( 1 + \frac{N}{2} \right)}{(1 + \frac{cN}{2})^{-\frac{2}{3}}} + \frac{1}{1-\alpha} \right), \qquad \text{(putting } M = \frac{N}{2} ) \tag{16}$$

$$= 2^{\frac{2}{3}} \left( 2 + cN \right)^{-\frac{2}{3}} \left( \alpha^{\frac{N}{2}} \left( 1 + \frac{N}{2} \right)^{\frac{5}{3}} + \frac{1}{1-\alpha} \right), \qquad \text{(as } c \leq 1) \tag{17}$$

$$\leq 2 \left( 1 + cN \right)^{-\frac{2}{3}} \left( \alpha^{\frac{N}{2}} \left( 1 + \frac{N}{2} \right)^{\frac{5}{3}} + \frac{1}{1-\alpha} \right), \tag{18}$$

$$\leq 2u_N^2 \left( \alpha^{\frac{N}{2}} \left( 1 + \frac{N}{2} \right)^{\frac{5}{3}} + \frac{1}{1-\alpha} \right), \qquad \text{(from definition of } u_N) \tag{19}$$

This implies

$$\sum_{k=0}^{N-1} \alpha^{N-k} \eta_k + \kappa \alpha^{N+1} \leq \eta_N \left( 2\alpha^{\frac{N}{2}} \left( 1 + \frac{N}{2} \right)^{\frac{5}{3}} + \frac{2}{1-\alpha} + \alpha^{N+1}(1 + cN)^{\frac{1}{3}} \frac{\kappa}{\eta_0} \right), \tag{20}$$

$$\leq c_\eta \eta_N, \tag{21}$$

$$\tag{22}$$

where $c_\eta = \max_{N \geq 0} \left( 2\alpha^{\frac{N}{2}} \left( 1 + \frac{N}{2} \right)^{\frac{5}{3}} + \frac{2}{1-\alpha} + \alpha^{N+1}(1 + cN)^{\frac{1}{3}} \frac{\kappa}{\eta_0} \right)$. This concludes the proof. $\square$

### A.3 SUFFICIENT INCREASE LEMMA

**Lemma 11.** *Let $\theta_k$ be the iterate obtained Algorithm 1 . Then,*

$$E[J^{\pi_{\theta_{k+1}}} - J^{\pi_{\theta_k}}] \geq \frac{\eta_k}{1-\gamma} \left[ E\|\nabla J^{\pi_{\theta_k}}\|_2^2 + E\langle D^{\pi_{\theta_k}} A^{\pi_{\theta_k}}, D^{\pi_{\theta_k}}(A_k - A^{\pi_{\theta_k}})\rangle - \frac{2L\eta_k}{(1-\gamma)^3} \right]$$

*where $E_{critic}[\cdot \mid \theta_m, m \leq k]$ is expectation over critic samples given the actor has already made updates from $\theta_0$ to $\theta_k$. Note that this is possible because samples are drawn independently for actor and critic in Algorithm 1.*

*Proof.* From the smoothness of the return, we have

$$E\left[\,J^{\pi_{\theta_{k+1}}}-J^{\pi_{\theta_k}}\,\right]\geq E\left[\,\langle\nabla J^{\pi_{\theta_k}},\theta_{k+1}-\theta_k\rangle-\frac{L}{2}\|\theta_{k+1}-\theta_k\|^2\,\right],$$

$$\geq E\left[\,\frac{\eta_k}{1-\gamma}\langle D^{\pi_{\theta_k}}A^{\pi_{\theta_k}},A_k\odot\mathbf{1}_k\rangle-\frac{2L\eta_k^2}{(1-\gamma)^4}\,\right],\qquad\text{(from update rule equation 6)}$$

$$\geq\frac{\eta_k}{1-\gamma}E\left[\,\langle D^{\pi_{\theta_k}}A^{\pi_{\theta_k}},D^{\pi_{\theta_k}}A_k\rangle\,\right]-\frac{2L\eta_k^2}{(1-\gamma)^4},\qquad(\text{ as }(s_k,a_k)\sim d^{\pi_{\theta_k}})$$

$$\geq\frac{\eta_k}{1-\gamma}\left[\,E\|\nabla J^{\pi_{\theta_k}}\|_2^2+E\langle D^{\pi_{\theta_k}}A^{\pi_{\theta_k}},D^{\pi_{\theta_k}}(A_k-A^{\pi_{\theta_k}})\rangle-\frac{2L\eta_k}{(1-\gamma)^3}\,\right],\qquad(\text{ as }\nabla J^{\pi_{\theta_k}}=D^{\pi_{\theta_k}}A^{\pi_{\theta_k}})$$

$\square$

## A.4 SUB-OPTIMALITY RECURSION

**Lemma 12.** *Taking $a_k=\frac{1-\gamma}{2}(J^*-J_k)$, we get the recursion*

$$a_k-a_{k+1}\geq\eta_k\left[\,c_1 a_k^2-c_2\eta_k\,\right],\tag{23}$$

*where $c_2:=\frac{c_q}{1-\gamma}+\frac{L}{(1-\gamma)^3}$ and $c_1:=\frac{c^2}{2SC_{PL}^2}$.*

*Proof.* From Sufficient Increase Lemma 6, we have

$$J_{k+1}-J_k\geq\frac{\eta_k}{1-\gamma}\left[\,E\|\nabla J^{\pi_{\theta_k}}\|_2^2-|E\langle D^{\pi_{\theta_k}}A^{\pi_{\theta_k}},D^{\pi_{\theta_k}}(A_k-A^{\pi_{\theta_k}})\rangle|-\frac{2L\eta_k}{(1-\gamma)^3}\,\right]$$

$$\geq\frac{\eta_k}{1-\gamma}\left[\,E\|\nabla J^{\pi_{\theta_k}}\|_2^2-c_q\eta_k-\frac{2L\eta_k}{(1-\gamma)^3}\,\right],\qquad\text{from Lemma 5}$$

$$\geq\frac{\eta_k}{1-\gamma}\left(\,E\frac{c^2(J^*-J^{\pi_{\theta_k}})^2}{SC_{PL}^2}-c_q\eta_k-\frac{2L\eta_k}{(1-\gamma)^3}\,\right),\qquad\text{(from GDL Lemma 3)}$$

$$\geq\frac{\eta_k}{1-\gamma}\left(\,\frac{c^2(J^*-J_k)^2}{SC_{PL}^2}-c_q\eta_k-\frac{2L\eta_k}{(1-\gamma)^3}\,\right),\qquad\text{(from Jenson's inequality)}$$

$$\geq\frac{\eta_k}{1-\gamma}\left(\,\frac{c^2(J^*-J_k)^2}{SC_{PL}^2}-\frac{2c_q\eta_k}{1-\gamma}-\frac{2L\eta_k}{(1-\gamma)^3}\,\right),\qquad\text{(from Lemma 5).}$$

$\square$

## A.5 CONVERGENCE OF ACTOR

**Lemma 13.** *The $a_k$ is upper bounded as*

$$a_k\leq\frac{1}{(1+c_6 k)^{\frac{1}{3}}},\qquad\forall k\geq 0.$$

*where $c_6=\frac{3c_1^2}{4c_2}$.*

*Proof.* Let $u_k=\left(\frac{1}{1+\frac{3c_1^2}{4c_2}k}\right)^{\frac{1}{3}}$ be the solution of the ode $\frac{du_k}{dk}=-\frac{c_1^2}{4c_2}u_k^4$ with $u_0=1$. The learning rate is chosen as $\eta_k=\frac{c_1}{2c_2}u_k^2$, implying $\lambda_0=\frac{c_1}{2c_2}$.

From suboptimality recursion equation 7, we have We have

$$a_{k+1}\leq a_k+c_2\eta_k^2-c_1 a_k^2$$

Let us define the function

$$h(x) := x + c_2\eta_k^2 - c_1\eta_k x^2.$$

Observe that function $h(x)$ is increasing for all $x \leq 1$. This is because the derivative of the function

$$\frac{dh(x)}{dx} = 1 - 2c_1\eta_k x \tag{24}$$

$$\geq 1 - 2c_1 x, \qquad (\text{as } \eta_k \leq 1) \tag{25}$$

$$\geq 1 - x, \qquad (\text{as } c_1 = \frac{c^2}{2SC_{PL}^2} \leq \frac{1}{2}), \tag{26}$$

$$\geq 0, \qquad \forall x \leq 1. \tag{27}$$

We will prove $a_k \leq u_k$ for $k \geq 0$ by induction arguments. By construction, we have $a_0 \leq u_0 = 1$. Assuming $a_k \leq u_k$, from the definition of $h$, we have

$$a_{k+1} \leq h(a_k), \tag{28}$$

$$\leq h(u_k), \qquad (\text{as } a_k \leq u_k \leq 1 \text{ and } h(x) \text{ is increasing for } x \leq 1) \tag{29}$$

$$= u_k + c_2\eta_k^2 - c_1\eta_k u_k^2, \qquad (\text{from definition of } h) \tag{30}$$

$$= u_k + c_2\lambda_0^2 u_k^4 - c_1\lambda_0 u_k^4, \qquad (\text{as } \lambda_k = \lambda_0 u_k^2) \tag{31}$$

$$= u_k - \frac{c_1^2}{4c_2} u_k^4, \qquad (\text{as } \lambda_0 = \frac{c_1}{2c_2}) \tag{32}$$

$$= u_k - \frac{c_1^2}{4c_2} \int_{x=k}^{k+1} u_k^4 dx, \qquad (\text{division of unity}) \tag{33}$$

$$\leq u_k - \frac{c_1^2}{4c_2} \int_{x=k}^{k+1} u_x^4 dx, \qquad (\text{as } u_x \text{ is a decreasing function}) \tag{34}$$

$$= u_{k+1}, \qquad (\text{ by definition } \frac{du_x}{dx} = -\frac{c_1^2}{4c_2} u_x^4). \tag{35}$$

Hence, from the induction arguments, we get, for all $k \geq 0$,

$$a_k \leq u_k = \frac{1}{(1 + \frac{3c_1^2}{4c_2}k)^{\frac{1}{3}}} \tag{36}$$

$$\leq \frac{1}{(1 + c_6 k)^{\frac{1}{3}}}, \tag{37}$$

where $c_6 = \frac{3c_1^2}{4c_2}$. $\qquad\square$

## A.6 EXPLORATION ASSUMPTION ITEM 1 EQUIVALENT MOMMENTUM BELLMAN OPERATOR

We define $P_\pi((s', a'), (s, a)) = P(s'|s, a)\pi(a'|s')$ and $D^\pi((s', a'), (s, a)) = \mathbf{1}\left((s', a') = (s, a)\right)$ $(1 - \gamma)\sum_{n=0}^{\infty} \gamma^n \mu^T (P^\pi)^n(s)$.

**Proposition 4.** $c_\gamma = \max_{\pi, Q} \frac{\|D^\pi(I - \gamma P_\pi)Q\|}{\|Q\|} \leq 1 + \gamma$.

*Proof.*

$$\|D^\pi(I - \gamma P_\pi)Q\| \le \|D^\pi Q\| + \gamma\|D^\pi P_\pi Q\| \tag{38}$$

$$\le \|Q\| + \gamma\|D^\pi P_\pi Q\|, \qquad (\text{as } \sum_{s,a}|D((s,a),(s,a))| = 1) \tag{39}$$

$$= \|Q\| + \gamma\sqrt{\sum_{s,a}\big(\ d(s,a)\langle P_\pi(\cdot|(s,a)),Q\rangle\ \big)^2}, \tag{40}$$

$$\le \|Q\| + \gamma\sqrt{\sum_{s,a}\big(\ d(s,a)\|P_\pi(\cdot|(s,a))\|\|Q\|\ \big)^2}, \tag{41}$$

$$\le \|Q\| + \gamma\|Q\|\sqrt{\sum_{s,a}\big(\ d(s,a)\ \big)^2\ \|P_\pi(\cdot|(s,a))\|^2}, \tag{42}$$

$$\le \|Q\| + \gamma\|Q\|\sqrt{\sum_{s,a} d(s,a)\|P_\pi(\cdot|(s,a))\|_1^2}, \tag{43}$$

$$= (1+\gamma)\|Q\|. \tag{44}$$

$\square$

**Proposition 5.** *For any policy $\pi$, let $Q_{n+1} = Q_n + \beta D^\pi\Big[\ R + \gamma P_\pi Q_n - Q_n\ \Big]$ then*

$$\|Q^\pi - Q_{n+1}\| \le \sqrt{1 - \frac{\lambda^2}{2}}\|Q^\pi - Q_n\|_2.$$

*Proof.*

$$U_n := D^\pi\Big[\ R - (I - \gamma P_\pi)Q_n\ \Big] \tag{45}$$

$$= D^\pi\Big[\ Q^\pi - \gamma P_\pi Q^\pi - (I - \gamma P_\pi)Q_n\ \Big], \qquad (\text{using } Q^\pi = R + \gamma P_\pi Q^\pi) \tag{46}$$

$$= D^\pi\big(\ I - \gamma P_\pi\ \big)\big(\ Q^\pi - Q_n\ \big) \tag{47}$$

Lets look at

$$\|Q^\pi - Q_{n+1}\|^2 = \|Q^\pi - Q_n - \beta U_n\|^2, \qquad (\text{definition of } Q_{n+1})$$

$$= \|Q^\pi - Q_n\|^2 + \beta^2\|U_n\|^2 - 2\beta\langle Q^\pi - Q_n, U_n\rangle$$

$$\le \|Q^\pi - Q_n\|^2 + \beta^2\|U_n\|^2 - 2\beta\lambda\|Q^\pi - Q_n\|^2, \qquad (\text{from Assumption 1})$$

$$\le (1 + 2\beta^2 - 2\beta\lambda)\|Q^\pi - Q_n\|_2^2, \qquad (\text{from Proposition 4})$$

$$\le (1 - \frac{\lambda^2}{2})\|Q^\pi - Q_n\|_2^2, \qquad (\text{taking } \beta = \frac{\lambda}{2}).$$

$\square$

