# OpenReview forum: "Improved Sample Complexity for  Global Convergence of Actor-Critic Algorithms"
_ICLR.cc/2025/Conference — ICLR 2025 Conference Withdrawn Submission_

### Official Review · Reviewer_N4Hb · 2024-10-15

**Soundness:** 2
**Presentation:** 3
**Contribution:** 2
**Rating:** 3
**Confidence:** 4

**Summary:**

This paper studies the convergence rates of the actor-critic algorithm for solving reinforcement learning problems. The authors establish an $O(\epsilon^{-3})$ sample complexity, claiming it improves the current state of the art.

**Strengths:**

The writing and organization of this paper are clear.

**Weaknesses:**

I have two major comments: (1) the results, and (2) the assumptions.

(1) It seems that the algorithm presented in this work is not the vanilla policy gradient but rather the natural policy gradient. Specifically, Algorithm 1, Line 3, appears to have the same update as in Lemma 15 of [1]. For the natural policy gradient, several results in the literature [2,3,4,5] have shown that it achieves geometric convergence when using increasing step sizes. Consequently, the natural actor-critic algorithm has an $O(\epsilon^{-2})$ sample complexity for global convergence.

Could the authors explain the main differences between the proposed algorithm and the natural policy gradient? If they are indeed the same algorithm, what are the main improvements of this work compared to those mentioned above?

>[1] Agarwal, A., Kakade, S. M., Lee, J. D., & Mahajan, G. (2021). On the theory of policy gradient methods: Optimality, approximation, and distribution shift. Journal of Machine Learning Research, 22(98), 1-76.

>[2] Lan, G. (2023). Policy mirror descent for reinforcement learning: Linear convergence, new sampling complexity, and generalized problem classes. Mathematical programming, 198(1), 1059-1106.

>[3] Xiao, L. (2022). On the convergence rates of policy gradient methods. Journal of Machine Learning Research, 23(282), 1-36.

>[4] Yuan, R., Du, S. S., Gower, R. M., Lazaric, A., & Xiao, L. (2022). Linear convergence of natural policy gradient methods with log-linear policies. arXiv preprint arXiv:2210.01400.

>[5] Chen, Z., & Maguluri, S. T. (2022, May). Sample complexity of policy-based methods under off-policy sampling and linear function approximation. In International Conference on Artificial Intelligence and Statistics (pp. 11195-11214). PMLR.

(2) The iid sampling from $d^{\pi_k}$ seems to be a strong assumption. One of the main challenges in analyzing the convergence rates of coupled stochastic iterative algorithms (such as the one in this work) is to deal with the noise. Realistically, one would implement the sampling process described in the top paragraph on page 7 while performing the update, making the noise sequence being a time-inhomogeneous Markov chain. The iid assumption seems to greatly simplify the analysis. Could the authors discuss relaxing the iid assumption to strengthen the practical relevance?

(3) It is not entirely clear to me why Assumption 1, Parts 2 and 3, are considered weaker than Part 1, as claimed by the authors. Could a proof be provided to show that Part 1 is automatically satisfied under either Part 2 or Part 3? An extended discussion on the relationships between these parts of the assumption and their implications for the analysis would be beneficial.

**Questions:**

See the weaknesses section.

---

### Official Review · Reviewer_xfMM · 2024-10-26

**Soundness:** 2
**Presentation:** 1
**Contribution:** 1
**Rating:** 3
**Confidence:** 3

**Summary:**

This paper shows an improvement from $O(\epsilon^{-4})$ to $O(\epsilon^{-3})$ for one kind of actor-critic algorithm.

**Strengths:**

This paper improve the finite-analysis bound from $O(\epsilon^{-4})$ to $O(\epsilon^{-3})$ for some kind of actor-critic algorithm.

**Weaknesses:**

First, I would like to point out that this paper is poorly written in several respects.

1. Inconsistent Notation: The notations are inconsistent throughout the paper. For instance, in line 660, there is a mix-up between capital
C and lowercase c.

2. Mathematical Errors: There are several mathematical errors. For example, in line 297, it seems that a square is missing.

3. Undefined Notations: Certain notations are used without definition, like $A_t$ in the algorithm.

4. Lemmas Relabeled in the Appendix: If the proofs refer to the same lemma, they should maintain consistent labeling throughout the paper. For instance, Lemma 4 in the main body is relabeled as Lemma 9 in the appendix.

It is the authors' responsibility to ensure that the paper is easy to follow and free from critical errors.The authors need to thoroughly review and revise the manuscript to address these issues.

Regarding the content, I have some additional concerns:

1. The algorithm analyzed in the paper is for a tabular AC method, as the critic update is entirely tabular. Comparing the results in the tabular case with those using a linear function approximator is not meaningful. I think this is the key detriment of this paper.

2. The algorithm assumes that the state-action pairs $(s,a)$ are sampled i.i.d. from $d^{\pi_{\theta_k}}$. This requires sampling the entire trajectory each time, which is impractical in real-world scenarios. The typical way of sampling $(s,a)$ is from a single trajectory. This is also a significant limitation.

**Questions:**

Please see my comments in Weakness section.

---

### Official Review · Reviewer_9xbc · 2024-11-01

**Soundness:** 2
**Presentation:** 1
**Contribution:** 2
**Rating:** 3
**Confidence:** 3

**Summary:**

This paper studied actor-critic(AC) algorithm in discounted reinforcement learning setting. The paper proposes a variant of AC where the critic uses a constant step size and that of actor's is diminishing in a given form. The paper claimed to have achieved $\mathcal{O}(\epsilon^{-3})$ for a $\epsilon$ sub-optimality gap target. Such a result reduces the gap from policy gradient with exact gradient. In addition, constant step size in critic component enhances the practicality of the algorithm.

**Strengths:**

* A claimed  $\mathcal{O}(\epsilon^{-3})$ sample complexity for  $\epsilon$ sub-optimality gap target.
* The use of constant critic step size enhances the practicality of the algorithm.

**Weaknesses:**

* Missing a body of bi-level actor-critic literature that is complementary to the line of approach in the current paper. For example,

[1] Xu, Tengyu, Zhe Wang, and Yingbin Liang. "Improving sample complexity bounds for (natural) actor-critic algorithms." Advances in Neural Information Processing Systems 33 (2020): 4358-4369.

[2] Chen, Z., Zhou, Y., Chen, R.R. and Zou, S., 2022, June. Sample and communication-efficient decentralized actor-critic algorithms with finite-time analysis. In International Conference on Machine Learning (pp. 3794-3834). PMLR.

[3] Hairi, F.N.U., Liu, J. and Lu, S., 2022. Finite-Time Convergence and Sample Complexity of Multi-Agent Actor-Critic Reinforcement Learning with Average Reward," in Proc. ICLR, Virtual Event, April 2022. Proc. ICLR.

* The presentation is subpar, which makes it difficult to get the gist of the analysis. In particular, there are two ideas seem to be critical in the analysis of the algorithm:
1. Slow change of policy, essentially decoupling actor and critic steps: However, this idea has not been provided with clear intuition and sufficient explanation. How to see the "slowness" in the given (actor) step size choice? "Slow" relative to which critical time scale? Furthermore, what is the intuition behind the decoupling given "slowness" not complete decoupling in terms of technical derivations?

2. How does "adversarial" component come to the analysis? What does this have to with "slow" change of policy?

* Rephrase two sentences between Line 71 - 73. It's not clear what the authors are trying to convey.

* Lots of typos and incomplete writings, a not comprehensive list:
1. Line 66: n -> In.
2. Caption under table 1, missing an "as" after such.
3. Last paragraph in section 1 is not complete.
4. Define $a^{*}$ in Line 207.
5. Line 268, the same expression appears twice.
6. Missing a square after the first inequality in Line 297.
7. Line 346, Lets -> Let's.
8. Line 380, "=" -> "-", also missing a coefficient factor.
9. Equation (4) and (5) appears the same.
10. Line 419 is missing an "is".
11. Line 463 is missing parentheses.

**Questions:**

See the weakness section. In addition,
1. Line 143 mentioned that (Xu 2020) requires additional computation and is highly challenging. However, it is not obvious why "highly challenging" or even "challenging" in the first place. Please elaborate.

2. Line 293 states that sample uniformly. Is it uniformly among state space $\mathcal{S}$ or collected samples $\{s_1,…,s_{i+1}\}$ so far ?

3. What does the last paragraph in Page 7 have to with item 3 of Assumption 1?

4. In the same paragraph, why it will lead to deterministic policy, why not a stochastic policy?

5. . In the same paragraph, why converging to deterministic policy potentially lead to better error?

---

### Official Review · Reviewer_5GV4 · 2024-11-04

**Soundness:** 4
**Presentation:** 4
**Contribution:** 2
**Rating:** 3
**Confidence:** 4

**Summary:**

In my opinion the paper is well written, well-structured and easy to read. However, given the current state of the literature, the results it has obtained have been obtained in a more general setting previously. Therefore I cannot recommend this work for acceptance.

**Strengths:**

The paper has very clear and readable presentation. Additionally the convergence methods seem novel.

**Weaknesses:**

The key shortcoming of this paper is that Gaur et al. (2024), has already proven the last iterate convergence with a sample complexity of $\epsilon^{-3}$. It does this for an infinite state and action space. Additionally, the work uses a decreasing actor step size and a constant critic step size. It also incorporates



References:

Mudit Gaur, Amrit Bedi, Di Wang, and Vaneet Aggarwal. Closing the gap: Achieving global convergence (Last iterate) of actor-critic under Markovian sampling with neural network parametrization. In Proceedings of the 41st International Conference on Machine Learning,

**Questions:**

Is there a way to extend this paper for an infinite state and action space? If that can be achieved, it would being parity between this work and  Gaur et al. (2024)  atleast in terms of the type of  MDP considered.  It might then be possible to argue that this work has some advantages over  Gaur et al. (2024) and may be fit for acceptance.

---

> ### Author Response · Authors · 2024-11-19
> **Author's response.**
>
> We thank the reviewer for the time spent reviewing this work.  We are glad that the reveiwer finds our methods to prove convergence novel.
>
> The work Gaur et al. 2024 (see its Algorithm 1) is very different than our vanila actor crtic. Although they use $O(\epsilon^{-3})$ new samples, but they are used multiple times using the buffer. As their Theorem 1 states, to achieve $\epsilon$-close policy, they do $O(\epsilon^{-1})$ iterates, in each iterate they generate $O(\epsilon^{-2})$ many new samples (hence the sample complexity of $O(\epsilon^{-3})$), however in every iterates, gradient estimation uses samples $O(\epsilon^{-4})$ times (from the buffer of $O(\epsilon^{-2})$).  To summarize, they require $O(\epsilon^{-3})$ many new samples, but samples are $O(\epsilon^{-5})$ times.
>
>
> While our algorithm use $O(\epsilon^{-3})$ new samples, each once, doesn't require memory to store sample buffer. This makes our algorithm very different and more efficient than Gaur et al 2024.

---

> ### Author Response · Authors · 2024-11-19
> **Author response 2**
>
> Extending this work to infinite state-space can be challenging and an interesting direction, we leave this for our future work.

---

### Note · Authors · 2024-11-19

**Comment:**

We thank the reviewers for their invaluable time spent reviewing this paper. We will gladly incorporate reviwers suggestion, especially on related work.  We withdraw this paper for the same.

**Withdrawal Confirmation:**

I have read and agree with the venue's withdrawal policy on behalf of myself and my co-authors.